# Relationship between Blood and Standard Biochemistry Levels with Periodontitis in Parkinson’s Disease Patients: Data from the NHANES 2011–2012

**DOI:** 10.3390/jpm10030069

**Published:** 2020-07-25

**Authors:** João Botelho, Patrícia Lyra, Luís Proença, Catarina Godinho, José João Mendes, Vanessa Machado

**Affiliations:** 1Clinical Research Unit (CRU), Centro de Investigação Interdisciplinar Egas Moniz (CiiEM), Instituto Universitário Egas Moniz, 2829-511 Almada, Portugal; patricialyra10@gmail.com (P.L.); cgodinho@egasmoniz.edu.pt (C.G.); jmendes@egasmoniz.edu.pt (J.J.M.); vmachado@egasmoniz.edu.pt (V.M.); 2Periodontology Department, Clinical Research Unit (CRU), CiiEM, Egas Moniz, CRL, 2829-511 Almada, Portugal; 3Quantitative Methods for Health Research Unit (MQIS), CiiEM, Egas Moniz, CRL, 2829-511 Almada, Portugal; lproenca@egasmoniz.edu.pt

**Keywords:** Parkinson’s disease, movement disorders, periodontitis, periodontal disease, hematologic tests, Vitamin D, oral health

## Abstract

People with Parkinson’s Disease (PD) are associated with the presence of periodontitis. We aimed to compare blood and standard biochemical surrogates of PD patients diagnosed with periodontitis with PD individuals without periodontitis. This retrospective cohort study used a sample from the National Health and Nutrition Examination Survey (NHANES) 2011–2012 that underwent periodontal diagnosis (*n* = 3669). PD participants were identified through specific PD reported medications. Periodontitis was defined according to the 2012 case definition, using periodontal examination data provided. Then, we compared blood levels and standard chemical laboratory profiles of PD patients according to the presence of periodontitis. Multivariable regression was used to explore this dataset and identify relevant variables towards the presence of periodontitis. According to the medication report, 37 participants were eligible, 29 were secure and 8 were unsecure PD medications regimens. Overall, PD cases with periodontitis presented increased levels of White Blood Cells (WBC) (*p* = 0.002), Basophils (*p* = 0.045) and Segmented neutrophils (*p* = 0.009), and also, lower levels of Total Bilirubin (*p* = 0.018). In the PD secure medication group, a significant difference was found for WBC (*p* = 0.002) and Segmented neutrophils (*p* = 0.002) for the periodontitis group. Further, WBC might be a discriminating factor towards periodontitis in the global sample. In the secure PD medication, we found gender, segmented neutrophils and Vitamin D2 to be potential discriminative variables towards periodontitis. Thus, periodontitis showed association with leukocyte levels alterations in PD patients, and therefore with potential systemic changes and predictive value. Furthermore, Vitamin D2 and gender showed to be associated with periodontitis in with secure medication for PD. Future studies should assess in more detail the potential systemic repercussion of the presence of periodontitis in PD patients.

## 1. Introduction

Periodontitis is a chronic inflammatory condition that targets the supporting structures of the teeth [1]. Dental plaque build-up, periodontopathic microbial specificity and the host immune response can collectively be considered as periodontitis etiology factors [2]. The presence of periodontal pockets, inflamed gingiva and alveolar bone loss in certain teeth or tooth sites clinically characterizes periodontitis, which can ultimately result in tooth loss [3]. Apart from its effects in the oral cavity, periodontitis repercussions also instigate slight systemic inflammation, which end up setting off or aggravating known chronic inflammatory diseases, such as cardiovascular diseases including high blood pressure [4], diabetes mellitus [5], rheumatoid arthritis [6] Furthermore and Alzheimer’s Disease [7,8,9]. Being one of the most prevalent conditions of the adult population worldwide, periodontitis frequency seems to be higher in the male gender while also increasing with age [10].

Parkinson’s disease (PD) is the second most frequent slowly progressive neurodegenerative condition that mostly affects the central nervous system [11]. Still with elusive causal factors to date, sporadic PD appears to be the conjugation of both genetic and environmental risk factors [12,13]. Being a heterogeneous disorder, PD clinical phenotype is characterized by a broad range of motor and non-motor symptoms, differing in onset age (which is most common at 65–70 years of age) and disease progression rates (faster in late-onset forms) [11,14]. PD classical motor features include resting tremor, muscular rigidity and bradykinesia, while a wide number of other motor and non-motor features contribute to PD disability and the deterioration of PD patients’ overall quality of life [15]. Dopaminergic drugs like levodopa and functional neurosurgery are still standard treatments, although tending to be a universal solution to a non-uniform disease [16,17]. PD increases with age and tends to affect more men than women [18,19,20]. In an overall aging population, PD cases are expected to duplicate in the next couple decades [17,21].

To date, the relationship between PD and periodontitis stands with PD associated motor impairments and cognitive decline that compromises oral hygiene habits and causes the deterioration of patient’s oral status [22]. Consequently, PD individuals seem to be at high risk of developing periodontitis [23,24,25,26,27]. Furthermore, it has been proposed that chronic neuroinflammation secondary to periodontitis systemic outcomes may lead to PD pathogenesis, initiation and progression [8,22,28]. However, to the best of our knowledge, the systemic repercussion of the presence of periodontitis on blood and biochemical surrogates on PD has never been investigated. Our hypothesis is that, as an infection, periodontitis in PD subjects might result in an increase of the leukocyte levels, though for the remaining levels this is still undetermined. 

Therefore, our primary aim was to compare blood and standard biochemical levels between periodontitis and non-periodontitis cases among Parkinson’s disease patients. Additionally, we aimed to evaluate if such changed biomarkers might contribute to predict the presence of periodontitis in PD patients.

## 2. Material and Methods

### 2.1. Population

The National Health and Nutrition Examination Survey (NHANES) 2011–2012 data is a representative multistage probability sample of non-institutionalized U.S. civilians survey to assess the health status through the Centers for Disease Control and Prevention (CDC) and Prevention National Center for Health Statistics (NCHS) website at https://www.cdc.gov/nchs/nhanes/index.htm. In this retrospective cohort study, periodontal examination data from the NHANES 2011–2012 was extracted. Our analysis deemed the following exclusion criteria: younger than 18 years of age; participants with medical exclusion from periodontal exam; non-complete periodontal status and edentulous patients. 

Oral health data collection protocols were approved by the CDC, NCHS Research Ethics Review Board, Atlanta (USA), and all participants gave written informed consent. All the examinations were conducted in a mobile examination center (see in detail in [29]). 

### 2.2. PD Definition

PD cases were confirmed through specific PD reported medications according to the NHANES database. In this way, patients reporting the use of Benztropine, Carbidopa, Levodopa, Ropinirole, Methyldopa, Entacapone, Cabergoline, Orphenadrine and Pramipexole were categorized as PD cases [30,31]. Then, we divided participants as PD cases according to secure PD medication (Benztropine, Carbidopa, Levodopa, Ropinirole, Methyldopa and Entacapone) [30,31] and unsecure PD medication (Cabergoline, Orphenadrine and Pramipexole) [30,31,32,33,34]. The unsecure PD group was defined because Cabergoline is used to treat high levels of prolactin hormone [32], Orphenadrine is used to treat muscle spasms in musculoskeletal conditions [33] and Pramipexole is also used to treat restless legs syndrome (RLS) [34].

### 2.3. Periodontal Clinical Examination

Periodontitis was defined as a minimum of 2 or more sites with clinical attachment loss (CAL) ≥ 3 mm and a periodontal pocket depth (PPD) ≥ 4 mm or one site with PPD ≥ 5 mm, as described by Eke et al. (2012). Data from the Periodontal Examination of NHANES 11–12 were treated through appropriate algorithms in Microsoft Office (MO) Excel to render the respective periodontal diagnosis. From this, we were able to render the number of missing teeth.

### 2.4. Demographics Characteristics

The demographic variables included were age, gender, smoking status and number of teeth. From the self-reported questionnaire, we categorized smoking status as current smoker (smoked more than 100 cigarettes and currently smoking), former smoker (smoked more than 100 cigarettes and currently not smoking) and non-smoker (never smoked). Diabetes mellitus was categorized as “yes” or “no” according to the self-reported questionnaire. High blood pressure was categorized according to previous medical confirmation of high blood pressure and if taking prescription for hypertension.

### 2.5. Blood and Standard Biochemical Profile Levels

Blood levels data included White Blood Cell (WBC) count (10^9^/L), percentage of Lymphocyte (%), percentage of Monocyte (%), percentage of Segmented Neutrophils (%), percentage of Eosinophils (%), percentage of Basophils (%), Lymphocyte (10^9^/L), Monocyte (10^9^/L), Segmented neutrophils (10^9^/L), Eosinophils (10^9^/L), Basophils (10^9^/L), Red Blood Cell (RBC) count (million cells/uL), Hemoglobin (g/dL), Hematocrit (%), Mean Cell Volume (MCV) (fL), Mean Cell Hemoglobin (MCH) (pg), Mean Cell Hemoglobin Concentration (MCHC) (g/dL), Red Cell Distribution (RCD) width (%), Platelet count (10^9^/L), Mean Platelet Volume (MPV) (fL).

For the Standard Biochemical Profile levels we included Albumin (g/dL), Alanine aminotransferase (ALT) (U/L), Aspartate aminotransferase (AST) (U/L), Alkaline phosphatase (AP) (U/L), Blood Urea Nitrogen (mg/dL), Total Calcium (mg/dL), Creatine Phosphokinase (CPK) (IU/L), Cholesterol (mg/dL), Bicarbonate (mmol/L), Creatinine (mg/dL), Gamma Glutamyl Transferase (GGT) (U/L), Glucose, Serum (mg/dL), Iron (refrigerated) (ug/dL), Lactate Dehydrogenase (LDH) (U/L), Phosphorus (mg/dL), Total Bilirubin (mg/dL), Total Protein (g/dL), Uric Acid (mg/dL), Sodium (mmol/L), Potassium (mmol/L), Chloride (mmol/L), Osmolality (mmol/Kg), Globulin (g/dL), Triglycerides (mg/dL), 25-hydroxyvitamin D2 (25OHD2) (nmol/L), 25-hydroxyvitamin D3 (25OHD3) (nmol/L).

### 2.6. Data Management and Analysis

Data were uploaded through SAS Universal Viewer for Windows and handled with MS Excel. For each periodontal case definition, specific MS Excel datasets were derived in order to formulate appropriate algorithms to define the periodontal status according to the case definition. Data analysis was performed using IBM SPSS Statistics version 25.0 for Windows (IBM CORP: ARMONK, NY, USA). Descriptive measures are reported through mean ± standard deviation (SD) for continuous variables, and number of cases (*n*), percentage (%) for categorical variables. The main outcome variable was the presence of periodontitis (P+ vs. P−). We compared baseline variables between periodontitis and non-periodontitis groups. Explicit comparison of mean values was performed by t-Student test when data assumptions for the application of this test were met (normality and homoscedasticity). Mann–Whitney test was used, as an alternative comparison technique, when those assumptions were not verified. To compare significant variables between the subgroups P(−) and P(+) we graphically computed the tendency of WBC, segmented neutrophils and basophils counts according to age using scatterplots from ggplot2 package for R version 4.0, and tendency was computed and fitted via ‘geom_smooth’. Then, we made regression analyses in the overall and only in secure PD cases. Preliminary analyses were performed using univariate models. Next, a multivariable model was constructed for the presence of periodontitis. Only variables showing a significance *p* ≤ 0.25 in the univariate model were included in the multivariable stepwise procedure. Predictor variables considered in this procedure were: gender (female as reference), WBC count (10^9^/L), Segmented neutrophils (10^9^/L) and 25-hydroxyvitamin D2 (25OHD2) (nmol/L). The contribution of each variable to the model was evaluated by Wald statistics. A multivariable stepwise adjusted logistic regression procedure was used to model the influence of the investigated factors towards the presence of periodontitis in PD patients. A significance level of 5% was set in all inferential analyses. 

## 3. Results

### 3.1. Population

From a total of 9756 participants, 3669 individuals had completed periodontal examination. From these, 37 (32 to 80 years old, 57.6 ± 14.6) participants were identified as taking PD medications, 29 secure (32 to 80 years old, 59.6 ± 14.7) and 8 unsecure PD (36 to 73 years old, 50.5 ± 12.5) medications regimens (Table 1). There were no age differences between PD cases with periodontitis (P+) and without periodontitis (P−). Males comprised 40.5% of the sample. The majority of subjects were non-smokers (55.9%). Diabetes and high blood pressure cases were evenly distributed. The number of missing teeth did not differ between PD cases with periodontitis and without periodontitis. 

### 3.2. Blood and Standard Biochemical Levels

Complete blood count with 5-part differential was used to compare blood levels of the periodontitis group defined by NHANES measures with the subset of subjects considered periodontally healthy (Table 2). Overall, periodontitis group presented increased levels of WBC (*p* = 0.002), Basophils (*p* = 0.045) and Segmented Neutrophils (*p* = 0.009), also displayed graphically (Figure 1). In the PD secure medication group, the same difference was found for WBC (*p* = 0.002) and Segmented Neutrophils (*p* = 0.002) for the periodontitis group (Figure 1).

Then, we investigated the standard biochemistry profile levels to investigate the systemic status of these participants according to the presence of periodontitis (Table 3). The only meaningful result was found in the global sample, where the periodontitis group presented lower levels of Total Bilirubin (*p* = 0.018).

### 3.3. Predictive Models of Periodontitis on PD Patients

In order to analyze which factors would discriminate the periodontitis presence, we performed multivariable stepwise regression analyses considering each factor. In the overall sample, blood WBC levels were consistently identified as a discriminative factor towards periodontitis (B = 0.773, *p* =0.025) (Table 4). Among the participants with secure PD medication, we found discriminative factors to be gender (male) (B = 5.126, *p* = 0.026), Segmented Neutrophils (B = 4.232, *p* = 0.027) and 25OHD2 (B = −0.127, *p* = 0.060). The second model evidenced an improved score for correct classification (89.7%).

## 4. Discussion

In the present representative study from the NHANES 2011–2012, periodontitis was associated with increased serum levels in PD patients. Therefore, our hypothesis was confirmed, in which leukocyte levels (WBC count, segmented neutrophils and basophils) and bilirubin were increased in periodontitis cases in this particular population. Furthermore, for the overall population WBC count showed potential predictive value towards periodontitis, while for secure PD medications gender, segmented neutrophils and 25OHD2 were the meaningful elements.

The link between periodontitis and leukocytosis is well documented [35,36,37,38,39,40]. This result is expected given the infectious nature of periodontitis where bacteria invade the periodontal tissues via the ulcerated epithelium, and leukocytes, in particular neutrophils, are triggered towards the periodontal injury [40,41,42]. Neutrophils had been associated with periodontitis pathogenesis [40,43,44] and were established as key players involved in many inflammatory chronic and aging-related diseases [44]. Neutrophils represent the vast majority (≥95%) of leukocytes recruited to the periodontal pocket [45]. Despite the homeostasis role of neutrophils in the healthy periodontium [3], they are impaired in periodontitis [1]. The chronic recruitment of excessive neutrophil, and therefore the increase of its serum counts, is learned as a consequence of the persisting microbial dysbiotic challenge [44]. The newness of this study is the likelihood of such parameters presenting predictive value towards periodontitis in PD cases, and future research is warranted to confirm this possibility.

Furthermore, male gender presented a higher risk to have periodontitis, this result being in line with previous reports that show males have a higher prevalence of periodontitis both in representative [10,46,47] and PD populations [23,24,25,26]. This result is of particular relevance because, in the same fashion as periodontitis, PD is more prevalent in men [18,19,20]. Additionally, the prevalence of periodontitis in this age-group is in line with previous studies developed in this region, where this age groups have high levels of periodontal disease [46,47,48].

Additionally, the presence of 25OHD2 in the predictive models is also in accordance with previous studies, where individuals with periodontitis were associated with lower levels of Vitamin D, compared to non-periodontitis [49,50,51,52,53]. Further, Vitamin D concentrations were associated with higher periodontal destruction, severe periodontitis stages and higher tooth loss [54,55,56,57,58]. Vitamin D also influences the immune response through the regulation of cathelicidin [59]. Interestingly, cathelicidin is an antimicrobial peptide produced by neutrophils and has been shown that dysregulated neutrophils in periodontitis lead to a low secretion of cathelicidin [60], though this should be further investigated. Therefore, Vitamin D levels may be an interesting clinical surrogate to consider in this link of periodontitis with PD, though it demands more studies to allow strong conclusions. However, we should carefully interpret these findings because of the lack of significance according to the periodontal status but its predictive value to infer periodontitis. 

The present report has limitations important to mention and discuss. Despite this sample deriving from a large representative U.S. population survey, the final number of included patients was small. However, this small number can be explained by the fact that PD affects 1% of individuals over 60 [11]. In our study, the overall prevalence of PD patients confirmed by medication represented 0.4% of the entire population and 1.0% of the sample that was examined for periodontitis. Thus, the sample size of this study limits the validity of these results and warrants future confirmation with prospective studies, since there are inherent biases in cross-sectional studies, such as selection bias. Notwithstanding, the identification of PD patients was also a limitation, since was based on the medication consumption present in the NHANES database with inherent selection bias. While for some medications this is somehow secure (Benztropine, Carbidopa, Levodopa, Ropinirole, Methyldopa and Entacapone) for others this is not the case (Cabergoline, Orphenadrine and Pramipexole) [30,31,32,33,34]. Yet, PD clinical diagnosis is even now considered to be speculative, since a definitive diagnosis always implies a post-mortem examination [13,61]. Another shortcoming is the medication itself since this survey was carried out in 2011–2012, and a large variation of medication gained therapeutic relevance in recent years. Furthermore, therapeutic adherence in PD is sub-optimal in a significant proportion of patients with PD [62], and we may have had a sample shortage due to this reason. Further, this approach does not deliver any causality rather an associative conclusion, and future studies should investigate in more depth how PD and periodontitis relate systemically, and if treating periodontitis might alleviate these elevated surrogates. Moreover, white blood cells and neutrophils were used as proxy of systemic inflammation and PISA as proxy to oral inflammation, though more evidence, such as immunohistochemistry staining of the periodontal tissues, indicating the infiltration level of neutrophils, monocytes and related white blood cells are warranted to further confirm our results to expand this matter. Lastly, the number of analyzed markers may be considered excessive, as future studies will narrow analysis to the most relevant measures. 

In spite of these limitations, this article has important strengths. Our report is the first to depict the potential effect of the presence of periodontitis on the systemic status of PD. Further, NHANES is prospectively a reliable source of data to determine associations as previously demonstrated [63], and public data bank analysis (such as NHANES) are key towards more comprehensive oral health studies. Furthermore, we were able to produce predictive estimates using serum surrogates, which may be clinically relevant for the multidisciplinary team of PD. These results underline the importance of oral health care and how it can become unbalanced with the progression of this neurodegenerative disease, and the importance of more studies to investigate the systemic influence of periodontitis on PD.

## 5. Conclusions

Periodontitis was associated with an increase of white blood cells count, segmented neutrophils and basophils in PD patients. Furthermore, white blood cells count, segmented neutrophils, Vitamin D2 and gender showed discriminatory value to predict the existence of periodontitis in PD cases. Future studies should assess in more detail the potential systemic repercussion of the presence of periodontitis in PD patients.

## Figures and Tables

**Figure 1 jpm-10-00069-f001:**
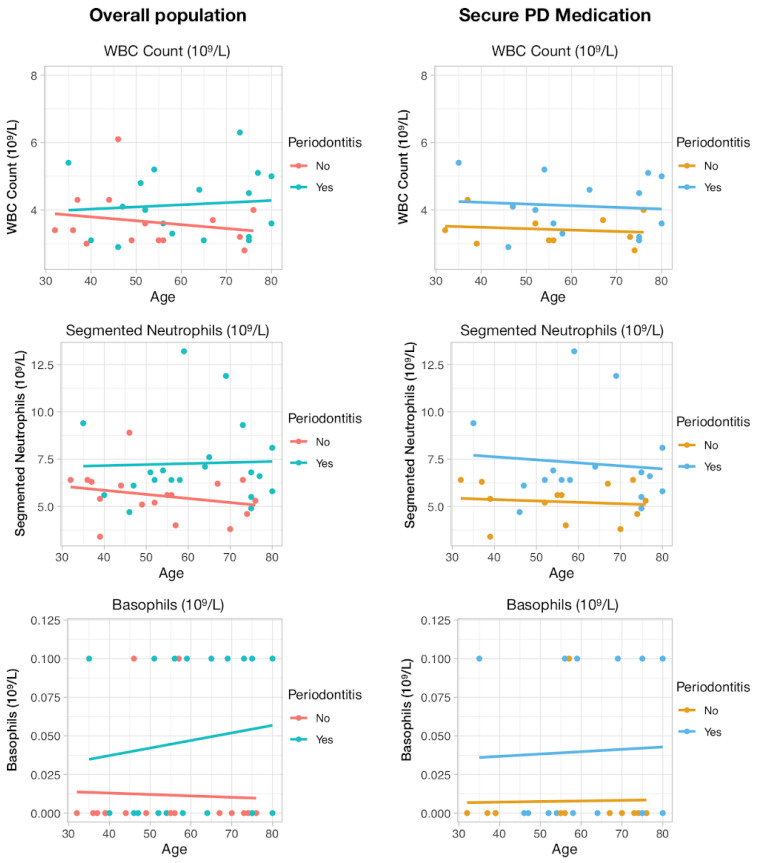
Comparison of WBC Count, Segmented neutrophils and Basophils serum levels between Periodontitis and no Periodontitis PD participants both in the overall sample and Secure PD medications. Lines represent graphically the tendency.

**Table 1 jpm-10-00069-t001:** Participants characteristics.

Variable	Global (*n* = 37)			Secure PD Medication (*n* = 29)		
	P(−)	P(+)	*p*-value ^†^	P(−)	P(+)	*p*-Value ^†^
**Age, mean (SD) (years)**	53.1 (14.6)	61.6 (13.8)	0.069	55.9 (15.4)	62.6 (13.8)	0.215
**Gender, n (%)**						
Female	12 (44.4)	10 (27.0)	0.204	10 (34.5)	7 (24.1)	0.071
Male	5 (13.5)	10 (27.0)		3 (10.3)	9 (31.0)	
**Smoking habits, n (%)**						
Never	11 (29.7)	12 (44.4)	0.668	7 (24.1)	9 (31.0)	0.588
Former	5 (13.5)	5 (13.5)		5 (17.2)	4 (13.8)	
Active	1 (2.7)	3 (8.1)		1 (3.4)	3 (10.3)	
**Diabetes Mellitus, n (%)**	3 (8.1)	2 (5.4)	0.498	3 (10.3)	2 (6.9)	0.453
**High Blood Pressure, mean (SD)**	10 (27.0)	10 (27.0)	0.591	9 (31.0)	9 (31.0)	0.474
**Missing Teeth, mean (SD)**	3.9 (5.8)	4.5 (4.3)	0.302	5.1 (6.2)	4.6 (4.6)	0.362

^†^ Chi-square test for categorical variables and Mann-Whitney test for continuous variables, *p* < 0.05. P(−)—No Periodontitis, P(+)—Periodontitis.

**Table 2 jpm-10-00069-t002:** Hematologic levels of PD patients with periodontitis and without periodontitis.

Variable	Global (*n* = 37)			Secure PD Medication (*n* = 29)		
	P(−)	P(+)	*p*-Value ^†^	P(−)	P(+)	*p*-Value ^†^
WBC count (10^9^/L)	5.57 (1.28)	7.28 (2.19)	0.002	5.25 (1.02)	7.26 (2.36)	0.002
Lymphocyte (%)	28.75 (6.19)	26.38 (6.91)	0.284	28.84 (6.47)	25.35 (5.87)	0.144
Monocyte (%)	7.36 (3)	7.01 (2.09)	0.988	7.65 (3.36)	6.89 (2.07)	0.812
Segmented neutrophils (%)	60.25 (7.48)	62.85 (8.34)	0.330	59.37 (7.53)	63.88 (6.82)	0.103
Eosinophils (%)	3.16 (2.2)	3.1 (1.5)	0.752	3.66 (2.27)	3.22 (1.56)	0.682
Basophils (%)	0.51 (0.36)	0.75 (0.95)	0.537	0.52 (0.38)	0.73 (1.05)	0.846
Lymphocyte (10^9^/L)	1.59 (0.49)	1.89 (0.66)	0.137	1.52 (0.5)	1.81 (0.63)	0.179
Monocyte (10^9^/L)	0.39 (0.17)	0.5 (0.19)	0.104	0.38 (0.19)	0.49 (0.19)	0.121
Segmented neutrophils (10^9^/L)	3.38 (0.98)	4.61 (1.66)	0.009	3.12 (0.72)	4.68 (1.73)	0.002
Eosinophils (10^9^/L)	0.18 (0.13)	0.23 (0.13)	0.297	0.21 (0.14)	0.23 (0.14)	0.682
Basophils (10^9^/L)	0.01 (0.03)	0.06 (0.08)	0.045	0.01 (0.03)	0.06 (0.08)	0.101
RBC count (million cells/uL)	4.37 (0.36)	4.45 (0.4)	0.528	4.28 (0.31)	4.48 (0.44)	0.186
Hemoglobin (g/dL)	13.64 (1.35)	13.92 (1.2)	0.519	13.35 (1.22)	14.02 (1.22)	0.155
Hematocrit (%)	39.85 (3.42)	40.48 (3.98)	0.614	39.13 (3.08)	40.9 (4.15)	0.213
MCV (fL)	91.34 (4.56)	91 (3.18)	0.794	91.48 (5.24)	91.33 (2.91)	0.922
MCH (pg)	31.23 (1.89)	31.29 (1.4)	0.919	31.17 (2.16)	31.32 (1.36)	0.822
MCHC (g/dL)	34.18 (0.84)	34.39 (1.01)	0.516	34.05 (0.89)	34.29 (1.1)	0.532
RCD width (%)	12.94 (1.05)	12.78 (0.69)	0.940	13.02 (1.2)	12.93 (0.5)	0.650
Platelet count (10^9^/L)	213.82 (41.73)	243.3 (86.16)	0.598	205.62 (38.2)	246.44 (91.76)	0.329
MPV (fL)	8.21 (1.19)	8.17 (0.8)	0.752	8.14 (1.27)	8.13 (0.79)	0.714

^†^ Mann-Whitney for continuous variables without normal distribution and *t*-test for continuous data with normal distribution, *p* < 0.05. Lymphocytes (%), Segmented neutrophils (%), RBC, Hemoglobin, Hematocrit, MCV and MCH were compared with *t*-test, and remaining with Mann-Whitney test. P(−)—No Periodontitis, P(+)—Periodontitis; WBC—White Blood Cells; RBC—Red Blood Cells; MCV—Mean Cell Volume; MCH—Mean Cell Hemoglobin; MCHC—Mean Cell Hemoglobin Concentration; RCD—Red Cell Distribution; MPV—Mean Platelet Volume.

**Table 3 jpm-10-00069-t003:** Standard biochemical levels of PD patients with periodontitis and without periodontitis.

Variable	Global (*n* = 37)			Secure PD Medication (*n* = 29)		
	P(−)	P(+)	*p*-Value ^†^	P(−)	P(+)	*p*-Value ^†^
Albumin (g/dL)	4.19 (0.31)	4.02 (0.97)	0.775	4.15 (0.32)	3.99 (1.09)	0.714
ALT (U/L)	21.53 (11.12)	20.6 (15.54)	0.517	21.77 (12.45)	21.25 (17.32)	0.682
AST (U/L)	24.53 (8.22)	22.95 (10.68)	0.821	25.38 (9.26)	23 (11.87)	0.650
AP (U/L)	76.41 (26.04)	75.45 (28.62)	0.916	81.92 (27.02)	75.19 (30.63)	0.540
Blood urea nitrogen (mg/dL)	14.12 (6.71)	14.00 (7.83)	0.916	14.92 (7.39)	14.06 (8.68)	0.812
Total calcium (mg/dL)	9.25 (0.39)	8.94 (2.13)	0.209	9.28 (0.42)	8.81 (2.37)	0.449
CPK (IU/L)	117.71 (69.34)	114.4 (71.15)	0.798	113.92 (67.13)	118.5 (73.93)	0.619
Cholesterol (mg/dL)	179.41 (39)	175.15 (48.95)	0.869	176.08 (42.41)	174.94 (54.9)	0.619
Bicarbonate (mmol/L)	25.06 (2.11)	23.1 (5.96)	0.232	25.38 (2.06)	22.69 (6.55)	0.092
Creatinine (mg/dL)	0.92 (0.29)	0.87 (0.3)	0.869	0.95 (0.31)	0.88 (0.33)	0.880
GGT (U/L)	20.76 (14.06)	26.9 (30.47)	0.684	21.92 (15.47)	29.19 (33.77)	0.812
Glucose, serum (mg/dL)	107.82 (51.97)	91.45 (27.09)	0.892	113.54 (58.4)	93.31 (29.78)	0.914
Iron, refrigerated (ug/dL)	89.12 (30.64)	74.65 (38.59)	0.080	85.15 (31.26)	72.88 (42.46)	0.170
LDH (U/L)	131.06 (24.8)	122.25 (36.4)	0.557	138.85 (21.86)	122.81 (39.98)	0.268
Phosphorus (mg/dL)	3.51 (0.51)	3.59 (0.9)	0.232	3.48 (0.57)	3.52 (0.99)	0.398
Total bilirubin (mg/dL)	0.68 (0.22)	0.5 (0.21)	0.016	0.64 (0.19)	0.51 (0.23)	0.110
Total Protein (g/dL)	6.92 (0.6)	6.63 (1.66)	0.940	6.82 (0.59)	6.61 (1.86)	0.475
Uric acid (mg/dL)	4.81 (1.39)	4.95 (1.65)	0.794	4.73 (1.31)	5.14 (1.74)	0.487
Sodium (mmol/L)	139.06 (1.92)	132.3 (31.19)	0.869	139.15 (1.68)	130.44 (34.83)	0.880
Potassium (mmol/L)	4.01 (0.27)	3.84 (0.98)	0.752	4.04 (0.25)	3.84 (1.1)	1.000
Chloride (mmol/L)	104.76 (2.61)	98.85 (23.45)	0.270	104.31 (2.56)	97.38 (26.16)	0.398
Osmolality (mmol/Kg)	278.65 (5.74)	264.75 (62.47)	0.619	279.38 (5.69)	261.31 (69.84)	0.779
Globulin (g/dL)	2.73 (0.45)	2.62 (0.77)	0.916	2.67 (0.45)	2.61 (0.86)	0.779
Triglycerides (mg/dL)	119.82 (86.92)	161.5 (109.11)	0.149	109.31 (68.72)	178.63 (115.18)	0.068
25OHD2+25OHD3 (nmol/L)	75.56 (21.29)	77.11 (34.24)	0.873	71.14 (18.69)	68.59 (30.07)	0.792
25OHD2 (nmol/L)	4.27 (8.53)	11.12 (33.31)	0.478	4.93 (9.73)	4.52 (12.07)	0.423
25OHD3 (nmol/L)	71.28 (23.24)	65.97 (32.46)	0.578	66.18 (21.07)	64.03 (29.9)	0.829
epi-25OHD3 (nmol/L)	3.47 (2.11)	3.68 (2.19)	0.557^†^	2.89 (1.28)	3.7 (2.36)	0.351

^†^ Mann–Whitney for continuous variables without normal distribution and *t*-test for continuous data with normal distribution, *p* < 0.05. Uric Acid and epi-25OHD3 were compared with Mann-Whitney test, and the remaining with *t*-test. P(−)—No Periodontitis, P(+)—Periodontitis; ALT—Alanine aminotransferase; AST—Aspartate aminotransferase; AP—Alkaline phosphatase; CPK—Creatine Phosphokinase; GGT—Gamma glutamyl transferase; LDH—Lactate dehydrogenase; 25OHD2—25-hydroxyvitamin D2; 25OHD3— 25-hydroxyvitamin D3.

**Table 4 jpm-10-00069-t004:** Final reduced logistic regression models for the overall population (*n* = 37) and patients with secure PD medication (*n* = 29).

	Crude Model				Adjusted Model			
	B	*p*-Value	Exp(B)	95% CI for Exp(B)	B	*p*-Value	Exp(B)	95% CI for Exp(B)
Model 1—Overall population (*n* = 37) ^1^								
WBC count (10^9^/L)	0.773	0.025	2.1	1.1–4.2	0.773	0.025	2.2	1.1–4.3
Model 2—Secure PD medication (*n* = 29) ^2^								
Gender (male)	5.064	0.024	158.3	2.0–12760.6	5.126	0.026	19.2	1.2–297.1
Segmented neutrophils (10^9^/L)	3.727	0.090	41.6	0.6–3069.7	4.232	0.027	14.2	1.57–128.8
25OHD2 (nmol/L)	−0.130	0.058	0.9	0.8–1.0	−0.127	0.060	0.9	0.8–1.0

^1^ R^2^(n) = 0.291, % correct classification = 75.0%. ^2^ R^2^(n) = 0.730, % correct classification = 89.7%. B—unstandardized regression coefficient; WBC—White Blood Cells.

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
