# Peer review of "Relationship between Blood and Standard Biochemistry Levels with Periodontitis in Parkinson’s Disease Patients: Data from the NHANES 2011–2012"

_jpm, 2020, doi:10.3390/jpm10030069_

Round 1
Reviewer 1 Report
This manuscript has been largely improved. I only have a few remaining comments.
- Page 2, line 76: “such changes” needs to be clarified. Do you mean “differences in biomarkers“?
- Page 4, line 145: IBM SPSS Statistics independent samples t-test procedure provides both pooled-variances and separate-variances t-test for equality of means. Thus, you can apply t-test even when the independent groups do not have equal variances (data do not comfort to the assumption of homogeneity of variances).
- Page 4, lines 148-149: Please specify the method used to compute the tendency. Did you fit linear regression lines?
- Please replace “multivariate” with “multivariable” through the article.
- Tables 1-3: Identify and name the significance tests in the footnote. Report median (and inter quartiles) for variables with skewed distribution when Mann-Whitney’s test was applied.
- Tables 1-3: You have well responded to my concern about data dredging. Please include a comment about this issue also in the discussion section.
- Table 4: In your response to reviewers, you have clearly clarified that only four variables were included in the regression modelling procedure. I would like to advise you to improve the reporting of the estimated regression models. A problem in in the interpretation of regression models is that the variables selected by stepwise manner into the model are often interpreted as the only important factors related to the response variable. Please do not report only the effect of WRC count in Model 1 or the effects of the three variables in Model2 selected by the software stepwise algorithm. More illustratively, report the crude (unadjusted) OR (with 95% CI) for each of the four variables. Then include a column where you report the adjusted OR (with CI) for each variable. This helps your readers to better interpret the findings.
Author Response
Esteemed Editorial Board,
We are pleased with the opportunity to revise and resubmit our manuscript “Relationship between blood and standard biochemistry levels with Periodontitis in Parkinson’s Disease patients: data from the NHANES 2011-2012” (Manuscript ID jpm-842196).
We are very grateful for the editor and reviewers’ comments, all have been considered and taken into profound consideration.
Manuscript changes are highlighted in the revised manuscript. Our point-by-point responses to all comments are detailed below. We hope the revised manuscript will better suit the Journal of Personalized Medicine. We are happy to consider further revisions and we thank you for your continued interest in our research.
REVIEWER 1:
This manuscript has been largely improved. I only have a few remaining comments.
1. Page 2, line 76: “such changes” needs to be clarified. Do you mean “differences in biomarkers“?
Answer: Yes, we meant “differences in biomarkers”. We have rephrased accordingly (Line 76).
2. Page 4, line 145: IBM SPSS Statistics independent samples t-test procedure provides both pooled-variances and separate-variances t-test for equality of means. Thus, you can apply t-test even when the independent groups do not have equal variances (data do not comfort to the assumption of homogeneity of variances).
Answer: We appreciate this remark.
3. Page 4, lines 148-149: Please specify the method used to compute the tendency. Did you fit linear regression lines?
Answer: We have provided more details on the computation of tendency and its fitness method: “and tendency was computed and fitted via ‘geom_smooth’” (Page 4, Lines 150-151).
4. Please replace “multivariate” with “multivariable” through the article.
Answer: We have replaced “multivariate” with “multivariable” through the article.
5. Tables 1-3: Identify and name the significance tests in the footnote. Report median (and inter quartiles) for variables with skewed distribution when Mann-Whitney’s test was applied.
Answer: We identified the significance tests in the footnote of Tables 1-3. Regarding the reporting of median (and inter quartiles) for variables with skewed distribution when Mann-Whitney’s test was applied, we believe that this is will contribute to the confusion of the reader, as the table will turn very hard to read. However, we would like to ask if not reporting would be acceptable to the reviewer, or as another option, to report as supplementary information for all variables. We look forward to have your feedback.
6. Tables 1-3: You have well responded to my concern about data dredging. Please include a comment about this issue also in the discussion section.
Answer: We added “Lastly, the number of analyzed markers may be considered excessive as future studies will narrow analysis to the most relevant measures.” (Page 9, Lines 265-267).
7. Table 4: In your response to reviewers, you have clearly clarified that only four variables were included in the regression modelling procedure. I would like to advise you to improve the reporting of the estimated regression models. A problem in the interpretation of regression models is that the variables selected by stepwise manner into the model are often interpreted as the only important factors related to the response variable. Please do not report only the effect of WRC count in Model 1 or the effects of the three variables in Model2 selected by the software stepwise algorithm. More illustratively, report the crude (unadjusted) OR (with 95% CI) for each of the four variables. Then include a column where you report the adjusted OR (with CI) for each variable. This helps your readers to better interpret the findings.
Answer: We have reported the crude (unadjusted) and adjusted models as requested, in Table 4.
Reviewer 2 Report
In this paper, the authors examined the Parkinson’s disease (PD) through the lens of periodontitis. They compared the blood parameters and standard biochemistry levels with periodontitis in PD patients and found the association of leukocyte levels in PD patients. However, the statistical analyses and biological examinations are not comprehensive. Here are some questions that need to be further addressed in order to make more convincing conclusions.
- The PD patients span a wide range of ages in the cohort studied here. Need to further dissect the age hierarchy from the young to the old PD patients, for example 35 - 45 years old, 46 - 55 years old, etc. Then divide the samples into periodontitis vs non-periodontitis groups in each hierarchy and compare the blood and biochemistry parameters.
- The mean age of periodontitis (-) is 53.1, while the mean age of periodontitis (+) is 61.6. There is a >8-year old gap between these two groups, which is too big. Because the older of the adults, the more overall chronic inflammation level tends to exist, which has been observed from the senescence field. Thus, the observed increase of inflammatory cells in the blood might not be ascribed to the PD in this case. It’s more convincing to make the age gap less than 3 between periodontitis (-) and periodontitis (+) PD patients.
- The overall white blood cells and neutrophils in the peripheral blood cannot precisely reflect the oral inflammation level. Need to add more evidence of local inflammation such as immunohistochemistry staining of the periodontal tissues, indicating the infiltration level of neutrophils, monocytes and related white blood cells.
- Is there any retrospective data showing any difference between PD patients with or without the uptake of VitD for a certain period of time? In fact, the majority of the examined biochemistry parameters were not significantly different between periodontitis (-) and periodontitis (+) PD patients, even the P-value of 25OHD2 is >0.05. If there are no observed clinical indications showing any involvement of VitD in PD patients with or without periodontitis, it’s hard to conclude that VitD could be a diagnostic predictor of periodontitis for PD patients.
Author Response
Esteemed Editorial Board,
We are pleased with the opportunity to revise and resubmit our manuscript “Relationship between blood and standard biochemistry levels with Periodontitis in Parkinson’s Disease patients: data from the NHANES 2011-2012” (Manuscript ID jpm-842196).
We are very grateful for the editor and reviewers’ comments, all have been considered and taken into profound consideration.
Manuscript changes are highlighted in the revised manuscript. Our point-by-point responses to all comments are detailed below. We hope the revised manuscript will better suit the Journal of Personalized Medicine. We are happy to consider further revisions and we thank you for your continued interest in our research.
In this paper, the authors examined the Parkinson’s disease (PD) through the lens of periodontitis. They compared the blood parameters and standard biochemistry levels with periodontitis in PD patients and found the association of leukocyte levels in PD patients. However, the statistical analyses and biological examinations are not comprehensive. Here are some questions that need to be further addressed in order to make more convincing conclusions.
1. The PD patients span a wide range of ages in the cohort studied here. Need to further dissect the age hierarchy from the young to the old PD patients, for example 35 - 45 years old, 46 - 55 years old, etc. Then divide the samples into periodontitis vs non-periodontitis groups in each hierarchy and compare the blood and biochemistry parameters.
Answer: We have read with great interest this commentary. However, the available data does not allow such exploration. We have 7 patients on 35-45, and 3 patients on 46-55, and so on. It is impracticable to perform this data dissection into age hierarchy. We hope to perform such analysis in a future and larger study, however at this point it is not possible, unfortunately. We hope you may understand this situation and we appreciate this valid and important suggestion.
2. The mean age of periodontitis (-) is 53.1, while the mean age of periodontitis (+) is 61.6. There is a >8-year old gap between these two groups, which is too big. Because the older of the adults, the more overall chronic inflammation level tends to exist, which has been observed from the senescence field. Thus, the observed increase of inflammatory cells in the blood might not be ascribed to the PD in this case. It’s more convincing to make the age gap less than 3 between periodontitis (-) and periodontitis (+) PD patients.
Answer: We acknowledge this reviewer point. However, there is no statistical difference between age comparing both groups. In this perspective, there is no strong reason to highlight this information as a limitation of the study. Nevertheless, if the editor finds it relevant for the readers best interest we are totally available to consider this information.
3. The overall white blood cells and neutrophils in the peripheral blood cannot precisely reflect the oral inflammation level. Need to add more evidence of local inflammation such as immunohistochemistry staining of the periodontal tissues, indicating the infiltration level of neutrophils, monocytes and related white blood cells.
Answer: We agree with the reviewers as we added “Moreover, white blood cells and neutrophils were used as proxy of systemic inflammation and PISA as proxy to oral inflammation, though more evidence such as immunohistochemistry staining of the periodontal tissues, indicating the infiltration level of neutrophils, monocytes and related white blood cells are warranting to expand this matter.” (Page 9, Lines 266-270).
4. Is there any retrospective data showing any difference between PD patients with or without the uptake of VitD for a certain period of time? In fact, the majority of the examined biochemistry parameters were not significantly different between periodontitis (-) and periodontitis (+) PD patients, even the P-value of 25OHD2 is >0.05. If there are no observed clinical indications showing any involvement of VitD in PD patients with or without periodontitis, it’s hard to conclude that VitD could be a diagnostic predictor of periodontitis for PD patients.
Answer: We appreciate this valid comment. Although the vitamin D biochemical parameters examined were not significant in both statistical analysis, in the logistic regression the 25OHD2 was an important value towards periodontitis. In this sense, we stayed: “However, we should carefully interpret these findings because of the lack of significance according to the periodontal status but its predictive value to infer periodontitis.” (Page 8, Lines 246-247)
Round 2
Reviewer 2 Report
The authors provided the revised version in a prompt manner. They rephrased properly in the manuscript. Even though for the IHC staining, they didn't perform the experiment, I believe the current version could address the main points of my comments. Thus, I would agree with the current version for publication.
This manuscript is a resubmission of an earlier submission. The following is a list of the peer review reports and author responses from that submission.
Round 1
Reviewer 1 Report
This manuscript aimed to assess the association between periodontitis and
Parkinson’s disease (PD) using data from the NHANES questionnaire. The authors found that gender, segmented neutrophils, and Vitamin D2 were significant variables in the study.
COMMENTS
There is limited scientific data exploring the association between oral health status among individuals with neurodegenerative diseases such as PD. This study adds to the literature useful observations on this association. A better understanding of this association would facilitate better patient management.
There are some minor issues in the manuscript that should be addressed:
- The NHANES was used in a recent study by Montero E. (J Clin Perio). This work demonstrated how information (age, gender, ethnicity….etc) from NHANES could be a reliable source of data to determine associations. Please include the article in the discussion section highlighting the importance of public data bank analysis (such as NHANES) on oral health studies.
- The manuscript would benefit from a more elaborated discussion about neutrophil biology/periodontium response for two main reasons:
- One of the main findings was the increase in neutrophils, and these PMNs are crucial components of the periodontal host response. Historically, neutrophils had been associated with periodontitis pathogenesis (classic example Loesche WJ et al 1988). In addition, recent studies had shown that neutrophils are important players implicated in many inflammatory chronic diseases and aging-related diseases (Periodontology2000 Hajishengalis, G).
- It is well known that Vitamin D also influences the immune response regulating cathelicidin, which is an antimicrobial peptide. Interestingly cathelicidin is produced by neutrophils.
Reviewer 2 Report
My comments to the authors are as follows:
- Please define the study design in the abstract and in the materials section.
- Page 2, line 78: Your study is not an experimental study. Replace ”Experimental Section” with “Materials and methods”.
- Page 2, lines 74-76: This section is confusing. Clarify “changes of serum markers” Have you measured changes in serum markers for each participant? I think your primary aim was to compare blood and standard biochemical levels between PD+ and PD- cases among Parkinson’s disease patients.
- The statistical intensity of this manuscript was close to the average of articles published in medical journals. However, the quality of statistical reporting and data presentation was weak, I scored 3 in a scale from 0 (poor) to 10 (very high). The statistical reporting should be improved.
- Page 3, line 135: Consider use of "Data management and analysis” " instead of "Data management, test methods and analysis”. Statistical tests are included in the data analysis.
- Page 3, lines 137-138: Help your readers and define “appropriate algorithms”.
- Page 4, lines 141-145: Define your main outcome variable (PD+ vs PD-) and the main explanatory variables. Identify the variables (and methods) for each analysis done in the study. Do not write “chi-square test for categorical variables” or “Mann-Whitney test for continuous variables”. Why have you used Mann-Whitney’s test? You report mean values and standard deviations for the quantitative variables. Mann-Whitney’s test compare median values between PD+ and PD- groups.
- Tables 1-4: I calculated that you reported a total of 110 p-values from your sample of 37 participants. This is a sign of data dredging with multiple comparisons (number of p-values than the number of subjects in the study).
- Figure 1: Have you fitted linear regression lines in PD+ and PD- sub-groups. Please describe this in the Data analysis sub-section of the Materials and Methods section.
- Page 4, chapter 3.3. Variables towards PD: This title is not appropriate.
- Page 4, lines 190-195. Your sample size is 37. How many variables you have included in the stepwise analysis? In regression modelling, the “one in ten” rule is a rule of thumb for how many explanatory parameters can be estimated from data when doing regression analysis while keeping the risk of overfitting low. The rule states that one explanatory variable can be studied for every ten cases.] For logistic regression the number of events is given by the size of the smallest of the outcome categories. I think you now have an overfitting issue in your analyses. This could explain why 25OHD2 was included and WBC count was not included in the Model 2 in Table 4.